# High-Density and Monodisperse Electrochemical Gold Nanoparticle Synthesis Utilizing the Properties of Boron-Doped Diamond Electrodes

**DOI:** 10.3390/nano12101741

**Published:** 2022-05-19

**Authors:** Kenshin Takemura, Wataru Iwasaki, Nobutomo Morita, Shinya Ohmagari

**Affiliations:** Sensing System Research Center, The National Institute of Advanced Industrial Science and Technology (AIST), Tosu 841-0052, Japan; wataru.iwasaki@aist.go.jp (W.I.); morita.nobutomo@aist.go.jp (N.M.); shinya.ohmagari@aist.go.jp (S.O.)

**Keywords:** boron-doped diamond electrodes, gold nanoparticles, electrochemical analysis, As(Ⅲ)

## Abstract

Owing to its simplicity and sensitivity, electrochemical analysis is of high significance in the detection of pollutants and highly toxic substances in the environment. In electrochemical analysis, the sensitivity of the sensor and reliability of the obtained signal are especially dependent on the electrode characteristics. Electrodes with a high density of nanomaterials, which exhibit excellent activity, are useful as sensor substrates for pollutant detection. However, the effective placement of high-density nanomaterials requires a high degree of control over the particle size, particle shape, and distance between the particles on the substrate. In this study, we exploited the properties of boron-doped diamond (BDD) electrodes, which have a wide potential window, and succeeded in coating a highly dense layer of gold nanoparticles (AuNPs) at high potential. The AuNP-modified BDD (AuNP-BDD) electrodes comprising less than 100 nm AuNPs at a density of 125 particles/µm were electrochemically synthesized over a short period of 30–60 s. The AuNP-BDD electrodes were applied for detecting arsenic, which is one of the most abundant elements, and exhibited a limit of detection of 0.473 ppb in solution.

## 1. Introduction

Arsenic, which ranks 20th among the most abundant elements in the Earth’s crust, is highly mobile in the environment. Therefore, it exists as dust in air and dissolves in rain water and other water sources, forming a cycle that sweeps over plants and animals [1]. In particular, the issue of groundwater contamination has become a major threat worldwide. The World Health Organization (WHO) standard for arsenic concentration in drinking water is 10 µg/L (ppb); however, in Bangladesh, where arsenic contamination is a significant problem, the arsenic concentration exceeds 50 ppb in several locations [2,3]. Studies on the relationship between long-term exposure to inorganic arsenic and human health have been conducted extensively, and a correlation has been identified between the incidence of lung cancer and arsenic concentrations in beverages [4,5]. Therefore, arsenic pollution has been recognized as a serious issue that causes public health problems in many areas where significantly high concentrations of arsenic are ingested through water and food [6].

The detection of arsenic in the environment is important for maintaining public health. Arsenic analysis is performed using a combination of high-precision pretreatments, such as high-pressure liquid chromatography and highly sensitive analyzers [7,8]. While high-precision analytical techniques enable highly quantitative and qualitative analyses, they involve multiple steps in sample preparation and handling [9,10]. They are significantly powerful in providing accurate analysis; however, to maintain public health, a method to conveniently monitor arsenic in water sources and construction sites and agricultural soils is required. Electrochemical analysis is one of the most suitable methods for onsite heavy metal analysis as it enables detection using a limited number of devices based on a simple process involving redox reactions of the analyte [11]. In particular, stripping voltammetry is an effective method for measuring trace amounts of arsenic by electrochemically depositing and then oxidizing it in a solution by reverse potential scanning [12].

In the electrochemical measurements of arsenic, the electrode material is one of the most important components that directly affect the sensitivity and reliability of detection [13]. Among the different electrodes, carbon electrodes are widely popular. Structurally, these electrode materials are roughly classified as amorphous carbon, diamond, graphite, fullerene, carbon nanotubes, and graphene [14]. Boron-doped diamond (BDD) electrodes have been extensively studied owing to their wide potential window and low background current. Moreover, they exhibit excellent chemical stability and physical strength, rendering them suitable electrodes for a wide range of applications. Consequently, they have been applied to the determination of heavy metals [15].

Gold nanoparticles (AuNPs) are the most studied noble metal nanoparticles and are used in a variety of applications [16,17]. The most notable characteristics of AuNPs are their optical and electrochemical properties [18,19,20]. Particularly, AuNPs can facilitate significant signal enhancement in electrochemical measurements in a short period [21]. This signal enhancement effect is achieved by the high electron transfer rate of the AuNPs at the electrode surface. Certain studies have demonstrated highly sensitive heavy metal measurements by modifying the electrodes by coating AuNPs [22,23,24]. Furthermore, it is known that controlling the density of AuNPs on the electrodes can improve the sensitivity of electrochemical sensors [25]. Several approaches exist for modifying AuNPs on electrodes. Chemical modification of synthesized AuNPs is the most common technique which enables the most uniform particle modification on an electrode [26,27]. Because chemical modifications are time-consuming to fabricate, the electrochemical method, which synthesizes AuNPs directly on the electrode within minutes, significantly reduces the time required to fabricate the electrode [28,29]. However, the uniformity of the synthesized particles is relatively poor. Therefore, the development of an improved electrochemical synthesis method would be beneficial.

In this study, AuNPs were densely packed on a BDD electrode and used in the detection of As(Ⅲ). The AuNPs were electrochemically synthesized, which was made possible by the BDD electrode. During electrochemical detection, the As(Ⅲ) ions present in the solution were electrodeposited as As(0) on the AuNPs. The electrodeposited As(0) was quickly released from the electrode at a specific applied voltage, resulting in a characteristic increase in current, as illustrated in Figure 1.

## 2. Materials and Methods

### 2.1. Chemical Reagents and Equipment

As(Ⅲ) standard solution, acetic acid, sodium acetate trihydrate, and iron, lead, copper standard solutions were purchased from FUJIFILM Wako Pure Chemical Corporation (Osaka, Japan). Potassium tetrachloroaurate (Ⅲ) was purchased from Sigma-Aldrich (St. Louis, MI, USA). Pt wire and Ag/AgCl electrode (BAS Inc., Tokyo, Japan) were used as the counter and reference electrodes for the electrochemical measurements, respectively.

Electrochemical measurements were performed using an electrochemical analyzer (ALS 832D, BAS Inc., Tokyo, Japan). Scanning electron microscopy (SEM) was performed using JSM-9100F (JEOL Ltd., Tokyo, Japan) and Ultra Plus (Carl Zeiss Microscopy GmbH). Energy-dispersive X-ray spectroscopy (EDX) was performed using a spectrometer manufactured by Oxford Instruments (Abingdon-on-Thames, UK) and XFlash 6-30 (Bruker Nano GmbH, Berlin, Germany). X-ray diffraction (XRD) of the materials was performed using SmartLab (Rigaku, Tokyo, Japan). X-ray photoelectron spectroscopy (XPS) of the electrode surface was performed using AXIS-165 (Shimadzu Corporation, Kyoto, Japan).

### 2.2. Fabrication of the BDD Electrode

Heavily boron-doped polycrystalline diamond films were prepared on Si(100) substrates by hot-filament chemical vapor deposition (CVD). Before the film growth, the Si substrates were pre-seeded with commercially available diamond nanopowder particles with a size of 4–6 nm (uDiamond^®^, Carbodenon Ltd., Vantaa, Finland), to facilitate the nucleation of diamond. Hydrogen, methane, and trimethylboron gases were fed into the CVD chamber at a total pressure of 1.3 kPa. The methane/hydrogen gas ratio was maintained at 3% during growth. The tungsten filament wires were then resistively heated using a DC power supply at a filament temperature of 2200 °C. The typical film thickness and boron concentration, as measured by secondary ion mass spectrometry, were 5 mm and >10^20^ cm^−3^, respectively. Details of the preparation conditions can be found in our previous report [30].

### 2.3. AuNP Coating on BDD

The AuNPs were coated on the BDD film via a short-time pulse-inducing method. The BDD was placed as the working electrode in a three-electrode system immersed in a 0.4% KAuCl_4₄_ solution used as an electrodeposition solvent. The AuNPs were formed on the BDD surface by applying a voltage of −1.8 V for 30 s using the chronoamperometry method without stirring. The AuNP-deposited BDD electrode was rinsed with ultrapure water to avoid the chemical reaction of residual gold chloride. Thereafter, the electrochemical properties of the AuNP-modified BDD (AuNP-BDD) were evaluated using voltammetry. The XRD, XPS, EDX, and SEM analyses were performed to characterize the surface-modified AuNPs.

### 2.4. Electrochemical Detection of As(Ⅲ)

Electrochemical detection of As(Ⅲ) was performed with the square-wave anodic stripping voltammetry (SWASV) technique. The arsenic ions in 0.1 M acetic acid buffer (AcONa; pH of 5 at 20 °C) were electrodeposited under an applied voltage of −0.7 V while stirring for 300 s. After the electrodeposition process, the stirring was stopped and the sample was maintained under the static potential condition of −0.7 V for 15 s. As the final step of the measurement, the voltage was swept from −0.3 to 0.3 V with a scan rate of 12 mV and an amplitude of 25.0 mV; the frequency was set at 45.0 Hz.

## 3. Results and Discussion

### 3.1. Optimization of AuNP Deposition and Characterization

In the electrochemical deposition of AuNPs, the reduction voltage of gold chloride was applied in accordance with the electrochemical behavior exhibited by the BDD under high voltage. The applied voltage conditions were determined based on the potential window of the BDD (Appendix A). The AuNPs were sparsely formed on the BDD under an applied voltage of −2.1 V. The number of particles per square micrometer area was 23 at the densest location and the particle size was nonuniform (Figure 1a). This is due to the electrolysis of water before the reduction of gold ions in the solution, which was caused by the application of a high negative voltage. Several bubbles adhered to the BDD immediately after the start of the AuNP synthesis, interfering with the contact between the gold ions and the electrode surface. The formation of a highly dense coating of AuNPs was confirmed at an applied voltage of −1.9 V, at which the adhesion of bubbles to the electrode was relatively slow (Figure 1b). The AuNPs with a high coating density and the best particle size uniformity were formed at an applied voltage of −1.8 V (Figure 1c). When the magnitude of the applied voltage was decreased, the particle density and uniformity were significantly reduced (Figure 1d). Here, −1.5 V is the voltage at which a rapid rise in current values was observed from the BDD potential window. In contrast to the preceding conditions, intense electrolysis of water did not occur. Moreover, the applied voltages of −1.2 and −1.0 V produced changes in parameters such as the particle size (Figure 1e,f). A comparison of the characteristics of the particles obtained under different voltages indicated that high particle density and size uniformity were maintained only under an applied voltage of −1.8 V. This is because of the rapid reduction of only gold ions in the rapid AuNP synthesis reaction. The wide potential window, which is a characteristic of the BDD, contributed significantly to the synthesis of uniform nanoparticles on the electrode surface at a high voltage in a short duration.

For the synthesis of AuNPs uniformly and quickly, it is necessary to control the particle growth. In the electrochemical synthesis process, the voltage application time has the greatest effect on particle growth. The synthesis of AuNPs with considerably uniform sizes and shapes was confirmed even over a short duration of voltage application of 30 s (Figure 2a). Monodispersed and dense coating of AuNPs was achieved when the voltage was applied for 60 s (Figure 2b). However, a decrease in the number of deposited particles was observed with an increase in the average particle size (Appendix A). When the duration of voltage application was increased to 90 s, particle enlargement due to the coalescence of the grown particles was observed (Figure 2c). Consequently, the number and uniformity of AuNPs grown on the BDD surface were significantly reduced. When the Au deposition time was increased further, localized unmodified sites were observed on the BDD surface (Figure 2d). This is presumably due to the generation of hydrogen gas on the electrode surface. The density and number of AuNPs on the BDD surface could be controlled to a certain extent by controlling the particle growth time. The particles found on the surface were confirmed to be AuNPs via EDX analysis (Appendix A). The crystal structure of the modified AuNPs was analyzed by XRD (Figure 2e). The characteristic peaks of the (111) and (220) crystallographic planes corresponding to the sp^3^ carbons of the BDD were observed at 2θ = 43.93° and 75.36°, respectively [31]. These peaks clearly indicate the characteristic crystal structure of BDD. The weak peaks at 2θ = 38.2°, 44.3°, 64.4°, and 78.2° correspond to the (111), (200), (220), and (221) planes of AuNPs, respectively. The increase in the intensity of the AuNP crystal peak with the particle growth time indicated that gold deposition occurred in the form of nanoparticles (Appendix A). The presence of gold was clearly confirmed by the XPS analysis of the AuNP-BDD obtained with a voltage application time of 30 s (Figure 2f).

### 3.2. Electrochemical Property and Optimization of the AuNP-BDD

The electrochemical properties of the AuNP-BDD prepared using the optimum applied voltage were evaluated. As shown in Figure 3a, the potential window of the AuNP-BDD in AcONa was generally higher than that of the bare BDD, and the generation of oxygen and hydrogen shifted to low voltage in the former case. Conversely, the background current was lower for the gold disk electrode, and oxygen and hydrogen generation occurred at higher voltages. When comparing the curves near the oxygen evolution potential, which was observed to be 1.2 V, the electrochemical properties of the AuNP-BDD electrode approached those of the gold disk electrode as the gold deposition time in electrode fabrication was increased. This result suggests that even when the Au deposition was carried out for 120 s, the coating consisted of AuNPs and did not form a thin film. However, as mentioned earlier, the uniformity of the particles grown on the surface could not be maintained when Au deposition was conducted for longer than 60 s. For more accurate electrochemical measurements, the electrodes obtained through AuNP synthesis for 30 s were used in the determination of As(Ⅲ). The difference in the coating density of AuNPs affects the detection sensitivity of the BDD electrode in the As(Ⅲ) analysis. To demonstrate this, arsenic was detected using the bare BDD electrode, and AuNP-BDD electrodes were obtained using different applied voltages for Au deposition (Figure 3b). The current value near 0 V, which is a characteristic electrochemical signal of arsenic, increased significantly for the AuNP-BDD electrodes with a high AuNP density fabricated at an applied voltage of −1.8 V.

On the other hand, the BDD electrode exhibited a low background current, but the As(Ⅲ) peak was only marginally visible. These results indicate that the coating density of the AuNPs on the BDD electrode surface strongly affects the sensitivity of As(Ⅲ) detection. The voltage applied during stripping also significantly affects the sensitivity of heavy metal ion detection [32]. The deposition voltage (E_dep_) should be at an overpotential that is at least as negative as the potential required to reduce As(Ⅲ) to As(0). If the potential at which hydrogen gas generation occurs is chosen, the detection sensitivity will be reduced because hydrogen absorption on the electrode interface interferes with the contact between the As(Ⅲ) and the electrode surface. Figure 3c shows that the most efficient arsenic stripping occurred at −0.7 V for the AuNP-BDD electrodes fabricated in this study. When E_dep_ was higher than −0.7 V, hydrogen absorption on the electrode was strong, decreasing the electrodeposition efficiency. Further, the electrodeposition time was optimized to maximize the detection sensitivity (Figure 3d). When compared to the peak current obtained without stripping, there was a significant increase in current at 0.1 V after only 10 s of stripping. However, when stripping was performed for more than 90 s, a new peak was observed at −0.12 V. This is most likely because some of the deposited As(0) changed their state owing to excessive and prolonged stripping. 

### 3.3. Electrochemical Detection of As(Ⅲ) Using AuNP-BDD

The SWASV measurement with the optimized parameters and measurement method resulted in a characteristic peak of As(Ⅲ) near 0.06 V (Figure 4a). A concentration-dependent increase in the current peak value of As(Ⅲ) was observed with an increase in the As(III) concentration from 1 to 150 ppb. The electrochemical detection of As(Ⅲ) using bare BDD was not possible at 10 ppb of As(III) in the solution (Appendix A). The modification of the BDD with AuNPs resulted in at least a 20-fold increase in sensitivity. When the background current was subtracted from the obtained peak current of the electrode, the plot of peak current vs. As(Ⅲ) concentration showed ideal linearity in the range of 1 to 10 ppb (Figure 4b). The limit of detection (LoD) was estimated to be 0.473 ppb, as calculated by adding the mean of the blank sample to 3.3 times the standard deviation (SD) of the blank sample (*n* = 10, SD: 3.3 × 10^−6^) [33]. A constant degree of linearity was maintained when plotted up to 150 ppb (Appendix A). This suggests that arsenic in solution is electrodeposited in a concentration-dependent manner in the optimized detection procedure. The AuNP-BDD could rapidly detect As(III) within 10 min of contact with the analyte solution, even at As(III) concentrations significantly lower than the environmental standard set by the WHO. The properties and performance of the AuNP-BDDs obtained so far were compared with those of the previously reported AuNP-modified electrodes (Table 1). The AuNP, synthesized at −1.8 V and taking advantage of the wide potential window of BDD, exhibited extremely uniform grain size. The density was more than twice that of the reported AuNP-BDD electrode. Compared to other reports on sensitivity, it does not demonstrate the highest sensitivity. However, the high sensitivity and the linear range of detection are excellent, and the low background current of BDD as an electrode to capture the current response when As(III) is removed from the AuNPs is a factor in achieving the present detection sensitivity.

The detection of As(III) in solutions containing other heavy metals was performed to simulate measurements in real environments such as soil samples (Figure 4c). The findings indicated that it is possible to measure the electrochemical signal of arsenic regardless of the presence of foreign substances, such as other heavy metal ions. To demonstrate the improvement of the As(Ⅲ) electrodeposition efficiency facilitated by the AuNPs, the measurement was stopped immediately after electrodeposition and the air-dried electrodes were subjected to EDX point analysis (Figure 4d and Table 2). The results of six measurements on AuNPs and three measurements on AuNP-free areas of the AuNP-BDD substrate confirmed that As(Ⅲ) was electrodeposited only on AuNPs. The EDX mapping conducted for the surface analysis at another area of the same electrode indicated that the deposited arsenic tended to be localized, especially on AuNPs (Appendix A–d). The small amount of arsenic detected in the EDX analysis is thought to be due to the rapid desorption of As(0) from the interface because the open-circuit voltage was applied for ~5 s when the sample was removed during voltammetry. Even under these experimental conditions, the results of As(Ⅲ) detection and elemental analysis suggest that AuNPs improved the electrodeposition efficiency of As(Ⅲ) at the electrode interface.

## 4. Conclusions

We fabricated BDD electrodes densely coated with uniform AuNPs by controlling the time and applied voltage for electrochemical AuNP synthesis. The dense and uniform AuNP coating on the BDD improved the heavy metal stripping efficiency of the electrode. The resulting LoD indicated the capability of the system in detecting As(Ⅲ) at concentrations below the environmental standard concentration and minimized the inter-electrode error. Hence, the standard deviation in the detection of As(Ⅲ) was low, indicating the satisfactory performance of the system as a sensor. The electrode for heavy metal determination developed in this study has considerable potential as a next-generation chemical sensor for the highly sensitive, simple onsite detection of toxic As(Ⅲ) ions. Further control of the material and nanoparticle ratio is expected to lead to improved sensitivity.

## Data Availability

All the data generated or analyzed during this study are included in this published article.

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
