# Peer review of "High-Density and Monodisperse Electrochemical Gold Nanoparticle Synthesis Utilizing the Properties of Boron-Doped Diamond Electrodes"

_nanomaterials, 2022, doi:10.3390/nano12101741_

Round 1

Reviewer 1 Report

 According to the title of the main document, this paper concerns the use of boron-doped electrodes covered by electrochemically deposited gold nanoparticles. On the other hand, the title of the supplementary material explains that the cited electrode is applied to the determination of As(III) in aqueous solution. Hence, first of all the Authors should pay attention to the coherence of main article and supplementary material. Also, several inaccuracies and no properly described aspects are present in the text and require a deep amendment. Therefore, in my opinion the article here reviewed can be considered for the acceptance after major revisions.
Here below remarks are detailed.

1.     Whichever title Authors choose the last sentence in the abstract should report what does the reported limit of detection refer to.
2.     Keywords: being As(III) the object of the electrochemical determination, it should be included in the keywords.
3.     The central-bottom image in scheme 1 is not clear: what does it mean? Maybe it is to explain that surface area and the potential window influence the features of the electrode but is really quite confusing. Moreover, what does “electrolytic concentration” refer to? Does it refer to As(III) concentration? As(III) should be more properly defined as analyte. “Electrolytic concentration” could refer to the solvent system, but an investigation on the varying of the concentration of sodium acetate is not reported: it could be interesting, but it is not investigated in the paper. 
4.     In materials and methods, “tetrachloridoaurate” is not correct, it should be modified as “tetrachloroaurate”.
5.     Fabrication of the BDD electrode: what is the diamond nanopowder particles size? was the diamond nanopowder commercially available?
6.     Electrochemical detection of As(III): scheme 1 reports a deposition potential value of -0.8 V for arsenic, not -0.7 V as in the text: which is the real value? Also, are the potential values reported against SCE, Ag/AgCl, NHE, or something else? Counter and reference electrodes should be specified.
7.     Why the potential sweep in the As determination starts from -0.3 V rather than from the deposition potential?
8.     Results: experimental conditions in figure S1 should be specified.
9.     Line 126: the potential of -2.1 V is probably referred to a first experiment, then the deposition of AuNPs was carried out under different applied potential. This evolution is clear to the reader when he/she reads the following, therefore the selection of the proper deposition conditions should be better described.
10.  Table S1: The standard deviation is reported with too much, not significant figures, and consequently the particle size value is expressed in a not-correct mode. And also the particle number/?m2 should be reported with the associated standard deviation.
11.  Figure 3: “sodium acetate” is not usually shortened as “NaAc” but preferably as “AcONa”, for instance. However, abbreviations should be generally stated in the experimental section.
12.  The x-axis in fig. 4b reports “Pb concentration”; furthermore, the inset 1-10 ppb does not seem corresponding to the same range in the whole figure: maybe does the main figure 4b refers to another analysis on Pb, rather than the analysis here reported on As?
13.  Also, what about “Fe ions” in figure 4? Is it supposed to be Fe(III)? what is the meaning of “heavy metal” in this context? if it is about the toxic and poisoning effect of these metals, Fe is not considered “heavy”.
14.  Do the Authors considered the possibility of As(V) as interference?
15.  Table 1: what do “Large/Small” mean? how much “large/small”?
16.  Figure S4: what do a,b,c,d represent in fig. S4? the authors should complete the caption to fig. S4 and also the comment in the main text.
17.  A comparison with other, previous literature results should be reported.
18.  Line 269: actually, standard deviation in the detection of As(III) is never reported in the paper. The Authors should discuss this aspect on figure 4b, and correct it with the right data (see comment 12).

Reviewer 2 Report

This paper demonstrated the fabrication of Au NP modified (electrochemically deposited) BDD electrode and its electrochemical detection of As(III). By the SEM measurement and EDX point analysis, the Au NP prepared on the BDD surface and improved As(III) deposition by Au NP was investigated. The electrochemical detection based on the SWASV and following calibration curve are presented. However, I cannot find what is new in this paper. When I searched about the As(III) sensor based on the Au NP modified BDD, I found a lot of papers (see below) about Au NP modified BDD sensor for As. For publication, the author should explain what is different and advanced in this paper comparing to the previous similar works.

Some related previous work:

Anodic stripping voltammetric determination of total arsenic using a gold nanoparticle-modified boron-doped diamond electrode on a paper-based device

Microchimica Acta volume 185, Article number: 324 (2018)

Modification of boron-doped diamond electrode with gold nanoparticles synthesized by allyl mercaptan as the capping agent for arsenic sensors

AIP Conference Proceedings 2242, 040031 (2020)

Anodic stripping voltammetry of inorganic species of As3+ and As5+ at gold-modified boron doped diamond electrodes

  1. J. Electroanal. Chem. 615, 145–153.

Development of a Method for Total Inorganic Arsenic Analysis Using Anodic Stripping Voltammetry and a Au-Coated, Diamond Thin-Film Electrode

Anal. Chem. 2007, 79, 6, 2412–2420.

Reviewer 3 Report

The manuscript reports the voltammetric sensor towards arsenic, using electrodeposited AuNPs for sensitisation of the measurement. Although this principle was investigated not only once before (e.g., in 2006 https://doi.org/10.2116/analsci.22.567), the manuscript show certain novelty which relies on elucidation of the sensitising mechanism, that is, comparison of suitability of different size AuNPs. Nevertheless, this issue should be addressed in more details, mainly in the point of view of EDX determination of amount of As absorbed on different types of BDD/AuNP electrodes.

Furthermore, before the acceptation, some other issues should be addressed:

Abstract – information about the type of analyte is missing.

Introduction – more information about deposition/in situ synthesis of AuNPs should be provided. There are many studies using electrodeposited AuNPs for voltammetric determination of As3+, they should be mentioned. Furthermore, the achieved results (LOD, sensitivity, linear range) should be compared with the previously achieved results, with the AuNP on diverse substrates.

AuNP synthesis – the initial solution was stirred or quiescent during the measurement?

The reported size and number of AuNPs should be compared with other reported methods. Apparently, the study pivots around the high density of small nanoparticles being the most efficient for the electrochemical determination of As, so it is worthy to mention also other studies, to discuss whether they achieved similar/smaller/larger number of AuNPs on the surface.

XRD data should be supplemented with the pristine BDD electrode. Furthermore, from the very small figure 2e the peaks for diamond (111) and Au (200) cannot be distinguished.

Inset in Figure 3a is hardly legible. Figure 3c – what were the deposition times?

The hydrogen evolution discussed at the beginning of page 6 – were the bubbles indeed confirmed visually? I mean from the voltammograms (fig. 3a) one would guess that HER would start at about – 1.2 V (significant increase of current, typical for HER), not -0.7 V

Figure 4c – what was concentration of the metals?

Fig 4d should be larger. Besides, the origin of „small“ nanoparticles should be explained – according to their composition they are not gold?

Fig S4 should also be larger.

Figure S3 – either depicts only anodic parts of cyclic voltammograms or it is a kind of linear sweep voltammetry.

Round 2

Reviewer 1 Report

The revised version of the paper has solved some of the faults of the previous one. However, some points still need a revision.

1) in my opinion, scheme 1 is still too “crowded”, and this aspect makes it no clear

2) "Reply: The Si substrate was seeded with commercially available nanodiamond dispersed solution, and the primal diamond grain sizes are 4–6 nm.": this information should be added to the text

3) Above all: all the values in table S1 are still expressed in a wrong way, they should be corrected. For instance: 42±5.9 is wrong, it should be 42±6; 54±12.8 is wrong, it should be 50±10; all values should be corrected, in both columns.

4) table 1: values reported in the last line are (still) wrong (see comment 3 above).

Reviewer 2 Report

revised well

Author Response

Thank you for your valuable time to review this issue. Your comments has helped us to improve the quality of the MS. 

Reviewer 3 Report

The revision was made in a way that addressed all comments, the manuscript can now be accepted. 

Author Response

Thank you for your valuable time to review this issue. Your comments have helped us to improve the quality of the MS.

Round 3

Reviewer 1 Report

The Authors properly revised all the recommended points, hence the paper is now publishable.